

# Elevational variation in morphology and biomass allocation in carpathian snowbell *Soldanella carpatica* (Primulaceae)

Piotr Kiełtyk

Institute of Biological Sciences, Cardinal Stefan Wyszyński University in Warsaw, Warsaw, Poland

## ABSTRACT

Plants growing along wide elevation gradients in mountains experience considerable variations in environmental factors that vary across elevations. The most pronounced elevational changes are in climate conditions with characteristic decrease in air temperature with an increase in elevation. Studying intraspecific elevational variations in plant morphological traits and biomass allocation gives opportunity to understand how plants adapted to steep environmental gradients that change with elevation and how they may respond to climate changes related to global warming. In this study, phenotypic variation of an alpine plant *Soldanella carpatica* Vierh. (Primulaceae) was investigated on 40 sites distributed continuously across a 1,480-m elevation gradient in the Tatra Mountains, Central Europe. Mixed-effects models, by which plant traits were fitted to elevation, revealed that on most part of the gradient total leaf mass, leaf size and scape height decreased gradually with an increase in elevation, whereas dry mass investment in roots and flowers as well as individual flower mass did not vary with elevation. Unexpectedly, in the uppermost part of the elevation gradient overall plant size, including both below-and aboveground plant parts, decreased rapidly causing abrupt plant miniaturization. Despite the plant miniaturization at the highest elevations, biomass partitioning traits changed gradually across the entire species elevation range, namely, the leaf mass fraction decreased continuously, whereas the flower mass fraction and the root:shoot ratio increased steadily from the lowest to the highest elevations. Observed variations in *S. carpatica* phenotypes are seen as structural adjustments to environmental changes across elevations that increase chances of plant survival and reproduction at different elevations. Moreover, results of the present study agreed with the observations that populations of species from the 'Soldanella' intrageneric group adapted to alpine and subnival zones still maintain typical 'Soldanella'-like appearance, despite considerable reduction in overall plant size.

## INTRODUCTION

Elevation gradients in mountains are characterised by rapid environmental changes over very short distances and increasing elevation in temperate seasonal zones is associated with changes in many factors crucial for plant growth, survival and reproduction (*Körner, 2021*). As elevation increases there is a decrease in temperature, atmospheric pressure, $CO_2$

Corresponding author
Piotr Kiełtyk, kieltyk@wp.pl

content and length of the growing season, whereas precipitation, solar radiation, as well as number of weather-related extreme events, for example, frequencies of strong winds and frost during growing season, increase with an increase in elevation (*Billings, 1974*; *Nagy & Grabherr, 2009*; *Takahashi, 2014*; *Körner, 2021*). Moreover, at high elevations soil depth, microbial activity, nutrients availability, soil resource competition and herbivory pressure are generally lower (*Rasmann et al., 2014*; *Körner, 2021*), whereas competition for pollinator services increases with elevation because the number and activity of pollinators declines in low temperature environment of high elevations (*Maad, Armbruster & Fenster, 2013*; *Zhao & Wang, 2015*; *Arroyo, Pacheco & Dudley, 2017*). Therefore, elevation is an important indicator of abiotic and biotic variations which influence plant performance and survival and may significantly alter functional characteristics of a plant species (*Hautier et al., 2009*; *Scheepens & Stöcklin, 2013*; *He et al., 2017*; *Trunschke & Stöklin, 2017*; *Kiełtyk, 2018*; *Miljković et al., 2019*; *Paudel et al., 2019*; *Kiełtyk, 2021a*; *Kiełtyk, 2021b*; *Rathee et al., 2021*; *Ahmad et al., 2023*).

The intraspecific elevational variation observed in many plants can result from their adaptive plasticity (*Dai et al., 2017*; *Hamann et al., 2017*; *Henn et al., 2018*) and/or their genetic adaptation to local conditions (*Byars, Papst & Hoffmann, 2007*; *Gonzalo-Turpin & Hazard, 2009*; *Hirano, Sakaguchi & Takahashi, 2017*; *Morente-López et al., 2020*) because the environment selects for the optimal phenotype adapted to local resource availability and habitat conditions (*Midolo et al., 2019*). To alleviate stress levels and increase the uptake of the limiting resources or reduce the need for these limiting resources, plants can respond to variation in the environment by adjusting their physiology, morphology and biomass allocation (*Nicotra et al., 2010*; *Freschet et al., 2018*). Hence, different elevational stresses across elevation gradients may select for different ecological strategies among individuals of the same species that are reflected in distinct patterns of morphological variation and biomass allocation along elevational gradients (*Seguí et al., 2018*).

Allocation of dry matter to different plant structures implies existence of investment trade-offs, because resources allocated to one organ or function are unavailable for other organs or functions (*Weiner et al., 2009*). For example, in herbaceous perennials much allocation to stem and leaves is advantageous for competition for photosynthetic light capture but less allocation to belowground parts is disadvantageous for water and nutrients acquisition as well as for assimilates storage (*Takahashi & Matsuki, 2017*). According to the optimality theory the relationship between dry mass allocation in below- and above-ground organs (the root:shoot ratio) varies with range of resource supply and plants allocate more of their dry mass to organs that capture the most limiting resource (*Reich et al., 2014*; *Blume-Werry et al., 2018*). It is predicted that plants growing in cold, high-elevation environments with low nitrogen availability should invest more of their biomass in below-ground organs, compared to plants growing at low elevations, where temperatures are higher and the soil is richer in nutrients (*Bloom, Chapin III & Mooney, 1985*; *McConnaughay & Coleman, 1999*). The resource-retentive or the stress-tolerant life strategy demonstrated by increased below ground allocation, particularly in storage organs (*Grime, 2001*), can increase the chance of plants survival and population persistence in abiotically stressful and unpredictable environments of high elevations, by provisioning

plants with stored reserves in particularly severe years when the amount of assimilated carbohydrates is not sufficient for year-to-year survival (*Chapin III, Schulze & Mooney, 1990*; *von Arx, Edwards & Dietz, 2006*; *Guo et al., 2012*). Conversely, the opposite pattern of allocation is expected for plants growing in denser and taller vegetation at low elevations, where climatic conditions are milder but competition for resources, particularly for light, is stronger (*Callaway et al., 2002*; *Read et al., 2014*). In such environment selection should promote more the resource-acquisitive strategy manifested in increased plant growth that results from higher allocation in aboveground parts, namely, stem and leaves (*Grime, 2001*). Other important life-history adjustment in plants growing along elevation gradients is associated with variations in flower size and floral allocation (*Arroyo, Pacheco & Aguilera, 2013*; *Dai et al., 2017*; *Basnett, Ganesan & Devy, 2019*). In entomophilous species, increasing flower size can be correlated with higher reproductive success. It is expected, that at high elevations where pollinators are scarce and competition for pollinator services increase (*Malo & Baonza, 2002*; *Maad, Armbruster & Fenster, 2013*; *Zhao & Wang, 2015*; *Arroyo, Pacheco & Dudley, 2017*) plants produce larger flowers because they are preferred by insect pollinators (*Malo & Baonza, 2002*; *Maad, Armbruster & Fenster, 2013*; *Totland, 2001*; *Totland, 2004*). And this, in turn, increases the chances of pollen deposition and producing viable seeds, and finally, achieving reproductive success (*Arroyo, Primack & Armesto, 1982*; *Ohara & Higashi, 1994*; *Bingham & Orthner, 1998*). However, other selective pressures can promote opposite adjustment in flower size across elevations. In fact, it has been suggested that producing smaller flowers as found in some species can be advantageous in resource limited and climatically severe environment of high elevations because smaller flowers have lower cost of their structural development and physiological maintenance (*Herrera, 2005*). Consequently, different selective pressures, abiotic and biotic, can influence plant phenotypes along elevation gradients and studying variations in plant morphology and dry mass allocation is important to understand environmental adaptations of plants (*Takahashi & Matsuki, 2017*). Accordingly, common trends in plant intraspecific variations with increasing elevation include, among others, reduced overall size, height and biomass (*Alexander et al., 2009*; *Zhu et al., 2010*; *Maad, Armbruster & Fenster, 2013*; *He et al., 2017*; *Paudel et al., 2019*; *Kiełtyk, 2021a*; *Kiełtyk, 2021b*), reduced leaf size (*Byars, Papst & Hoffmann, 2007*; *Kiełtyk, 2018*; *Miljković et al., 2019*; *Paudel et al., 2019*) and leaf mass (*Kiełtyk, 2021a*; *Kiełtyk, 2021b*), lower number of flowers (*Kelly, 1998*; *Baret et al., 2004*; *Šťastná, Klimešowá & Doležal, 2012*; *Maad, Armbruster & Fenster, 2013*; *Gabel, Sattler & Reisch, 2017*), increased flower size and mass (*Kudo & Molau, 1999*; *Malo & Baonza, 2002*; *Herrera, 2005*; *Maad, Armbruster & Fenster, 2013*; *Kiełtyk, 2021b*), and increased seed mass (*Kudo & Molau, 1999*; *Alexander et al., 2009*; *Wu et al., 2011*; *Qi et al., 2015*). However, the opposite patterns with decreases in flower size (*Totland, 2001*; *Zhao & Wang, 2015*; *Hattori et al., 2016*) and seed mass (*Totland, 2004*; *Wirth et al., 2010*; *Gabel, Sattler & Reisch, 2017*) with an increase in elevation have also been reported for some species. Furthermore, in the Asteraceae family a shift in floral allocation patterns was observed in some species despite steady gross dry mass investment in flowers across elevation gradients. Namely, plants growing at high elevations had smaller numbers of larger flower heads with larger numbers of flowers as compared to low-elevation plants that had larger numbers but

smaller flower heads with smaller numbers of flowers (*Takahashi & Matsuki, 2017*; *Kiełtyk, 2021b*). Overall, the variety of elevational adjustments reported for different plant species suggests, that there are no general patterns of plant intraspecific elevational variations, but plant phenotypic responses to elevation may be species specific and context dependent (*e.g., Olejniczak et al., 2018*).

Patterns of intraspecific phenotypic variation in plants growing along elevation gradients in mountains are drawing increased research interest (*e.g., Halbritter et al., 2018*; *Olejniczak et al., 2018*; *Basnett, Ganesan & Devy, 2019*; *Rixen et al., 2022*; *Khatri et al., 2023*; *Spitzer et al., 2023*; *García-Rodríguez et al., 2024*). Studying variations in morphology and biomass allocation among plant organs across elevations provides relevant insights for understanding plant adaptive responses to biotic and abiotic variations along steep environmental gradients. Moreover, knowledge on life-history adjustments in steep climatic gradients as found along mountain slopes contributes to our understanding of how plants may cope with present and predicted future climate changes (*e.g., Frei et al., 2014*; *Pfennigwerth, Bailey & Schweitzer, 2017*; *Blume-Werry et al., 2018*; *Midolo et al., 2019*; *Rixen et al., 2022*). However, knowledge on intraspecific elevational variation in plant morphology most often is obtained from limited number of elevational sites. Accordingly, there are many studies in which plant morphological variation was analysed on two elevational sites, *i.e.,* on low and high elevation sites (*e.g., Kudo & Molau, 1999*; *Totland, 2001*; *Arroyo, Pacheco & Aguilera, 2013*; *Trunschke & Stöklin, 2017*), three sites, *i.e.,* low, medium and high (*e.g., Blionis & Vokou, 2002*; *Dai et al., 2017*; *Ahmad et al., 2018*; *Seguí et al., 2018*; *Vélez-Mora, Trigueros-Alatorre & Quintana-Ascencio, 2021*) or four elevational sites (*e.g., Milla et al., 2009*; *Sakurai & Takahashi, 2017*; *He et al., 2017*; *Rathee et al., 2021*). Furthermore, some number of studies examined trait variations on several to dozen (5–13) elevational sites (*e.g., Malo & Baonza, 2002*; *Herrera, 2005*; *Alexander et al., 2009*; *Šťastná, Klimešowá & Doležal, 2012*; *Zhao & Wang, 2015*; *Hattori et al., 2016*; *Takahashi & Matsuki, 2017*; *Miljković et al., 2019*; *Paudel et al., 2019*; *Cruz-Maldonado et al., 2021*; *García-Rodríguez et al., 2024*). However, studies of plant within-species phenotypic variation performed for greater number of elevational sites are scarce (*e.g., Kelly, 1998*—15 sites; *Maad, Armbruster & Fenster, 2013*—17 sites; *Kiełtyk, 2021b*—20 sites; *Khatri et al., 2023*—26 sites; *Kiełtyk, 2021a*—34 sites; *Kiełtyk, 2018*—47 sites). However, while analyses based on low number of elevational points can demonstrate and statistically test existence of trait variations across elevations, analyses performed for greater number of elevational sites, in which plant traits are modelled by continuous elevation variable, have an advantages of revealing shape and trajectory of elevational traits variations across plants vertical ranges. Accordingly, it has been recently postulated that researches on intraspecific trait variation of mountain plants representing different evolutionary lineages and growth forms, possibly based on large number of elevational sites, are of high priority to enhance our knowledge on plant adaptive responses across elevation gradients (*Rixen et al., 2022*).

In this study variation in a mountain herb *Soldanella carpatica* Vierh. (Primulaceae) was investigated on 40 sites distributed approximately evenly across a 1,480 m elevation gradient in the Tatra Mts., Central Europe. Phylogenetic relationships, phenotype evolution and ecology of species of the mountain genus *Soldanella* has been recently extensively studied

(*Zhang, Comes & Kadereit, 2001*; *Zhang & Kadereit, 2002*; *Steffen & Kadereit, 2014*; *Bellino et al., 2015*; *Štubňová et al., 2017*; *Slovák et al., 2023*; *Rurik et al., 2024*), and the significance of elevation as a key environmental gradient that has driven differentiation between montane and alpine species of the genus has been acknowledged. However, the selective pressure that have led to evolution of high elevation congenerics within the *Soldanella* genus, *i.e., S. minima*, *S. pusilla* and *S. alpina* (*Zhang, Comes & Kadereit, 2001*), should also operate at species level, resulting in emergence of phenotypic variations across elevations analogous to these observed between low and high elevation taxa of the genus. Indeed, it has been recently found that elevation has driven parallel adaptation to elevation and niche differentiation within populations of *Arabidopsis arenosa* (*Kolář et al., 2016*; *Knotek et al., 2020*; *Bohutínská et al., 2021*; *Wos et al., 2022*) and *Heliosperma pusillum* (*Szukala et al., 2023a*; *Szukala et al., 2023b*). Accordingly, low and high elevation populations of these species investigated in different mountain regions exhibited converged genetic and morphological responses to elevation, what indicates that elevation gradients can exert strong directional selection on plant adaptive traits leading to emergence of local ecotypes (*Wos et al., 2022*; *Szukala et al., 2023a*). Although morphological and genetic differentiation of the taxa within *Soldanella* genus has been studied, including their elevational diversification, up to date there are not any studies focused on elevational variation at intraspecific level within the genus. *S. carpatica* is a particularly good model species for analysing plant phenotypic responses to elevation due to its large elevational range in the region spanning from lower montane forests at the Tatra Mts. foothills to subnival zone on mountain ridges. Moreover, this species is widespread in the study region what allows for sampling plants on large number of elevational sites and model its phenotypic variations across continuous elevation gradient. The aim of this study was to reveal the overall effects of elevation on variations in set of fitness-related vegetative and reproductive traits of *S. carpatica*. In particular, the following questions were addressed: (1) does plant size expressed by plant dry mass and leaf size decrease with increasing elevation?, (2) do total flower mass as well as individual flower mass change with elevation?, (3) does increasing elevation is positively correlated with the root: shoot ratio?

## MATERIALS & METHODS

### Study species

The genus *Soldanella* (snowbells, Primulaceae) includes up to 24 mountain-dwelling taxa (18 species and six subspecies) occurring in the European Alpine System (*Zhang, Comes & Kadereit, 2001*; *Štubňová et al., 2017*; *Slovák et al., 2023*). Snowbells are small, caespitose or stoloniferous growth evergreen perennials, with entire coriaceous leaves and scapose, not leafy inflorescence bearing funnel-shaped or campanulate flowers. Showy, mostly blue-violet or pink-violet flowers appear immediately after snow melts (*Pawłowska, 1972*; *Štubňová et al., 2017*). Within the genus *Soldanella* two sections were previously distinguished based on floral and fruit morphology, namely, section *Soldanella* Pawłowska and section *Tubiflores* Borbás (*Pawłowska, 1963*; *Pawłowska, 1972*; *Zhang & Kadereit, 2002*), however, recent molecular investigations indicate that these phenotypic sections do not

reflect true phylogenetic relationships within the genus (*Zhang, Comes & Kadereit, 2001*; *Steffen & Kadereit, 2014*; *Slovák et al., 2023*; *Rurik et al., 2024*). Evolutionary history of this genus has been extensively affected by hybridization and introgression detected in almost all snowbell species (*Slovák et al., 2023*; *Rurik et al., 2024*). The 'Soldanella' group includes taxa 13–35 cm tall with several violet to violet-blue funnel-shaped flowers per scape, and capsules that open with ten teeth. This group comprises the majority of *Soldanella* species that inhabit predominately habitats from deciduous, mixed, or coniferous forests of the montane to the shrubs of subalpine belt but also able to occupy open habitats in the alpine zone (*Zhang, Comes & Kadereit, 2001*). In contrast, the 'Tubiflores' group comprises high alpine taxa up to 10 cm tall, characterized by overall dwarfism, single white to pink bell-shaped flowers, and capsules opening with five teeth. Taxa of this group occupy open habitats in the alpine and subnival zones (*Zhang, Comes & Kadereit, 2001*).

The study species, *S. carpatica*, member of the 'Soldanella' group, is a species endemic to the Western Carpathians and widespread in the Tatra Mts. The scape of *S. carpatica* is erect, (3)5–15(20) cm tall, not leafy, with (1)2–5- violet insect-pollinated flowers gathered at the top inflorescence (*Pawłowska, 1972*). Suborbicular leaves are gathered in a basal rosette (Fig. 1). The leaf blade is 8–50 mm wide, dark green, usually violet beneath with basal sinus narrow and shallow (*Pawłowska, 1963*). The species blooms from April to September and fruits from May to October (*Zhang & Kadereit, 2002*). *S. carpatica* has a broad edaphic tolerance and grows on both base-rich and base-poor soils developed from variety of bedrock (*Zhang, Comes & Kadereit, 2001*). In the Tatra Mts. this species occurs from lower montane forest (in Polish Tatra Mts. up to 1,200 m a.s.l.), upper montane forest (1,200–1,550 m a.s.l.), dwarf pine (=subalpine) belt (1,550–1,800 m a.s.l.), alpine belt (1,800–2,250 m a.s.l.) to subnival belt (above 2,250 m a.s.l.) (*Pawłowska, 1963*; *Mirek, 1996*) reaching elevation of 2,400 m a.s.l. (*Valachovič et al., 2019*).

Besides of *S. carpatica* there are also two other taxa of the 'Soldanella' group reported from the Tatra Mts., namely, *S. marmarossiensis* agg. and *S. montana*. Both of these taxa, however, are not as frequent as *S. carpatica* and they grow in the Tatra Mts. at lower mountain elevations. *S. marmarossiensis* agg. and *S. montana* are montane elements with preference for spruce forest, albeit *S. marmarossiensis* can also penetrate the alpine belt (*Valachovič et al., 2019*). It is noteworthy that due to the lack of species from 'Tubiflores' group in the Western Carpathians, only *S. carpatica* filled all the ecological niches above timberline, in the alpine and subnival belts (*Valachovič et al., 2019*). Indeed, in recent revision of *Soldanella* species occurrence in different vegetation types in Carpathians and the adjacent mountains *Valachovič et al. (2019)* found *S. carpatica* presence in 99% of the relevés from alpine grasslands in the Western Carpathians.

**Study area**

The investigated elevational gradient was located in the Polish part of the Tatra Mountains, Western Carpathians (Fig. 2), within the protected area of the Tatra National Park (study permission of the Tatra National Park: Bot/380 DBN.503/28/18). Being the highest mountain range of the Carpathians (2,655 m a.s.l., Gerlach peak in the Slovak part of the mountains), the Tatra Mts. are the only alpine mountain system with a well-developed

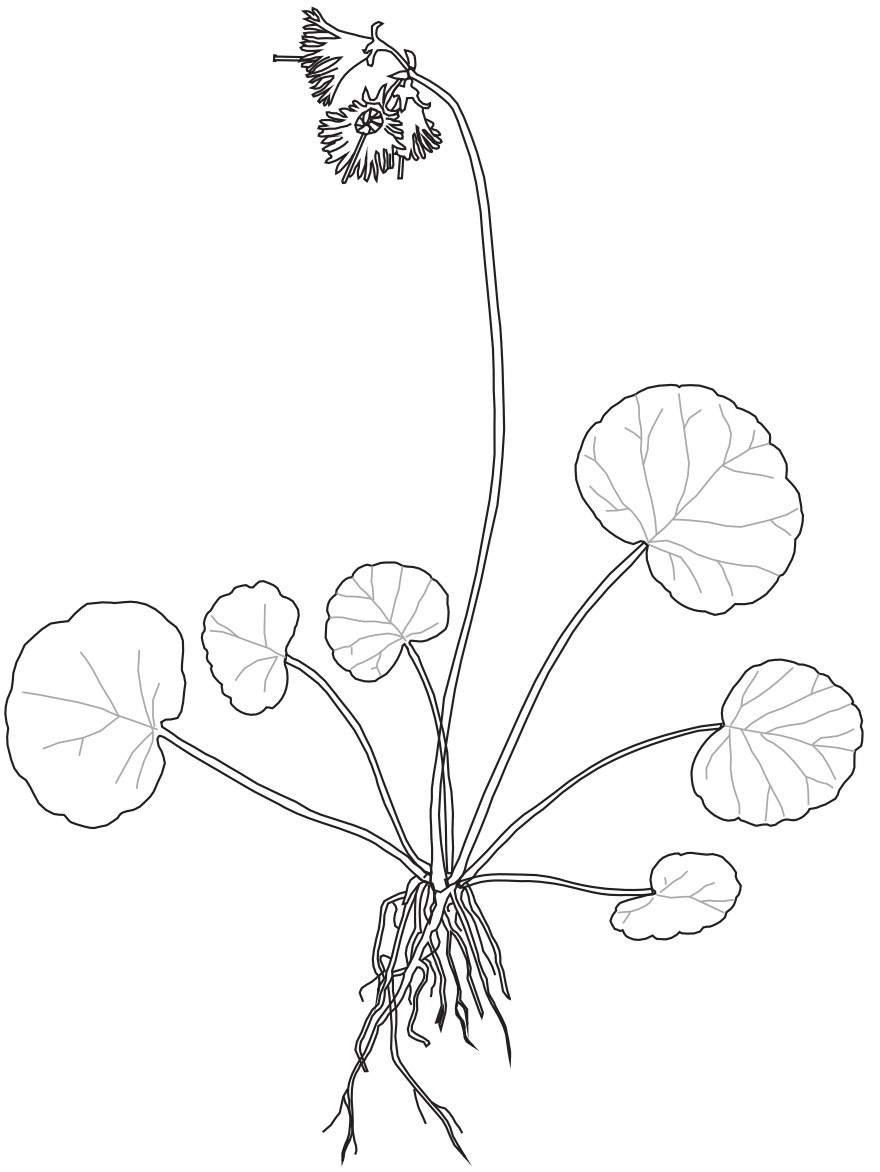

**Figure 1** *Soldanella carpatica* **habit.**

subnival zone between the Alps and the Caucasus (*Mirek, 1996*). The main ridge of the Tatra Mts. forms a natural border between Poland and Slovakia. The Polish side of the mountains is dominated by valleys and north-facing slopes and ranges from ca. 900 m to 2,499 m a.s.l. (Rysy peak).The mean annual temperature decreases from ca. +6 °C at the foothills to −4 °C at the highest peaks, whereas snow cover lasts for about 100 days at the foothills up to 290 days on mountain tops (*Hess, 1996*). Precipitation increases with elevation and the mean annual sum of precipitation averages 1,138 mm at an elevation of 844 m a.s.l. (the weather station in Zakopane town) and 1,876 mm at an elevation of 1,991 m a.s.l. (the weather station on Kasprowy Wierch peak) (*Hess, 1996*).
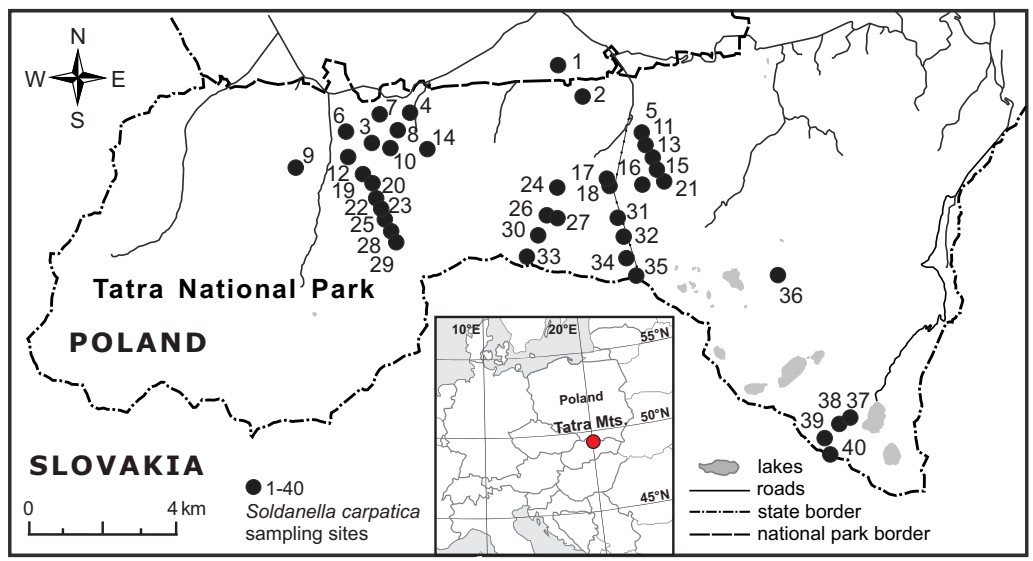

**Figure 2** Location of the 40 elevational sites sampled for *Soldanella carpatica* in the Polish Tatra Mts., Western Carpathians.

## Field sampling and traits measurements

A total of 383 flowering plants of *S. carpatica* were collected in 2018, from the end of April to the beginning of July, from 40 sites distributed from 890 m a.s.l. to 2,370 m a.s.l. (Table 1). The elevation at each site was determined in the field using a GPS receiver with built-in barometric altimeter (Garmin GPS MAP 76s, Olathe, USA). An attempt was made to sample entire elevational range of the species in the area and ensure that the sampled sites were distributed approximately evenly along the species elevational range. At each site, 9–10 plants in blossom peak with single stalk were sampled and carefully excavated with roots. To ensure that the sampled plants were separate genets, the minimal distance between the sampled plants was 2 m. Sampled individuals were well-developed and did not show restriction in growth and reproductive function as well as symptoms of fungal infection nor herbivory damages. Roots were roughly cleaned of soil and plants were preserved as dry material for laboratory analyses.

Eight traits were measured on dry plant material and further three traits were calculated as mass ratio traits; details on their measurements are presented in Table 2. Scape height was assessed as a distance between the plant base just below rosette leaves and the top of inflorescence; during the measurement the scape was straightened. Leaf blade diameter, the trait that assess size of photosynthetically active leaf part, was averaged for two largest plant leaves; on each leaf blade width was measured in two perpendicular directions and averaged per plant. In the next step plants were separated into four fractions, namely roots, scape, leaves and flowers, and final cleaning of roots was performed over a 0.6 mm mesh sieve under running water. Next, all the fractions were dried for 48 h at 80 °C in a laboratory drying oven with natural air circulation (Pol-Eko-Aparatura SLN 240, Wodzisław Śląski, Poland) to obtain the dry matter content (*Pérez-Harguindeguy et al.,*

**Table 1 Study sites of *Soldanella carpatica* in the Polish Tatra Mountains.** Geographic coordinates were determined with a WGS84 geodetic system.

| Site | Elevation (m a.s.l.) | Latitude (N) | Longitude (E) | Date |
|---|---|---|---|---|
| 1 | 890 | 49°16′54.8″ | 19°57′27.2″ | 2018-05-02 |
| 2 | 920 | 49°16′44.8″ | 19°57′43.6″ | 2018-05-02 |
| 3 | 948 | 49°15′47.7″ | 19°52′14.2″ | 2018-04-28 |
| 4 | 1,025 | 49°16′18.2″ | 19°52′49.6″ | 2018-04-29 |
| 5 | 1,075 | 49°15′52.4″ | 19°58′55.8″ | 2018-05-03 |
| 6 | 1,080 | 49°15′40.0″ | 19°53′07.7″ | 2018-04-28 |
| 7 | 1,083 | 49°16′10.1″ | 19°52′48.6″ | 2018-04-29 |
| 8 | 1,099 | 49°16′10.7″ | 19°52′52.3″ | 2018-04-29 |
| 9 | 1,120 | 49°15′21.0″ | 19°51′25.4″ | 2018-04-29 |
| 10 | 1,170 | 49°16′03.6″ | 19°52′55.2″ | 2018-04-28 |
| 11 | 1,175 | 49°15′27.8″ | 19°59′31.8″ | 2018-05-29 |
| 12 | 1,240 | 49°15′16.6″ | 19°52′51.9″ | 2018-05-01 |
| 13 | 1,242 | 49°15′24.9″ | 19°59′37.2″ | 2018-05-03 |
| 14 | 1,255 | 49°15′56.5″ | 19°53′08.3″ | 2018-04-28 |
| 15 | 1,280 | 49°15′17.8″ | 19°59′38.9″ | 2018-05-03 |
| 16 | 1,315 | 49°15′07.7″ | 19°59′41.4″ | 2018-05-29 |
| 17 | 1,340 | 49°15′01.3″ | 19°58′29.3″ | 2018-04-30 |
| 18 | 1,359 | 49°15′05.0″ | 19°58′29.8″ | 2018-04-30 |
| 19 | 1,390 | 49°15′02.6″ | 19°53′04.8″ | 2018-05-01 |
| 20 | 1,448 | 49°15′00.2″ | 19°53′12.6″ | 2018-05-01 |
| 21 | 1,450 | 49°15′05.8″ | 20°00′03.2″ | 2018-05-29 |
| 22 | 1,483 | 49°14′51.8″ | 19°53′13.8″ | 2018-05-01 |
| 23 | 1,520 | 49°14′49.3″ | 19°53′11.3″ | 2018-05-01 |
| 24 | 1,580 | 49°14′19.2″ | 19°56′34.6″ | 2018-05-28 |
| 25 | 1,618 | 49°14′40.2″ | 19°53′17.4″ | 2018-05-26 |
| 26 | 1,635 | 49°14′16.4″ | 19°56′30.3″ | 2018-05-28 |
| 27 | 1,650 | 49°14′15.5″ | 19°56′32.5″ | 2018-06-14 |
| 28 | 1,700 | 49°14′31.4″ | 19°53′25.9″ | 2018-05-26 |
| 29 | 1,746 | 49°14′25.9″ | 19°53′28.5″ | 2018-05-26 |
| 30 | 1,750 | 49°14′09.5″ | 19°56′23.7″ | 2018-06-14 |
| 31 | 1,800 | 49°14′13.9″ | 19°58′41.0″ | 2018-05-25 |
| 32 | 1,860 | 49°14′07.8″ | 19°58′43.4″ | 2018-05-25 |
| 33 | 1,863 | 49°14′03.5″ | 19°56′18.6″ | 2018-06-14 |
| 34 | 1,925 | 49°13′59.3″ | 19°58′49.9″ | 2018-05-28 |
| 35 | 1,980 | 49°13′55.1″ | 19°58′53.5″ | 2018-05-25 |
| 36 | 2,085 | 49°13′38.7″ | 20°01′51.8″ | 2018-07-10 |
| 37 | 2,175 | 49°11′04.1″ | 20°04′10.1″ | 2018-07-07 |
| 38 | 2,214 | 49°11′02.4″ | 20°04′08.0″ | 2018-07-07 |
| 39 | 2,280 | 49°11′02.7″ | 20°04′01.2″ | 2018-07-07 |
| 40 | 2,370 | 49°10′58.1″ | 20°04′01.0″ | 2018-07-07 |

**Table 2** *Soldanella carpatica* **traits used in the study.**

| Trait | Measurement details | Accuracy/ significant digits |
|---|---|---|
| **Aboveground plant mass**—total dry mass of the aboveground plant parts. Calculated as sum of leaves, stalk and flowers masses weighed separately. | Plant parts weighed on an analytical balance after drying for 48 h at 80 °C. | 0.0001 g |
| **Root mass**—dry mass of all plant roots. | Roots washed on the 0.6 mm sieve under running water and weighed on an analytical balance after drying for 48 h at 80 °C. | 0.0001 g |
| **Total leaf mass**—total dry mass of all plant leaves. | Weighed on an analytical balance after drying for 48 h at 80 °C. | 0.0001 g |
| **Scape mass**—dry mass of plant scape. | Weighed on an analytical balance after drying for 48 h at 80 °C. | 0.0001 g |
| **Total flower mass**—total dry mass of all plant flowers. | Weighed on an analytical balance after drying for 48 h at 80 °C. | 0.00001 g |
| **Individual flower mass**—total flower mass divided by number of flowers in the inflorescence. | – | 0.00001 g |
| **Scape height**—measured from the plant base to the top of the inflorescence. | Straightened scape was measured with a ruler on herbarium specimens. | 1 mm |
| **Leaf blade diameter**—mean value of the two perpendicular blade width measurements. This trait was measured and averaged for two largest plant leaf blades. | Measured with a digital calliper on herbarium specimens. | 0.1 mm |
| **Root: Shoot ratio**—root mass divided by plant aboveground mass. | – | – |
| **Leaf mass fraction**—total leaf mass divided by total plant mass. | – | – |
| **Flower aboveground mass fraction**—total flower mass divided by plant aboveground mass. | – | – |

2013) by weighing on an analytical balance (Radwag AS 60/220.X2 PLUS, Radom, Poland). All weight measurements were carried out immediately after the samples were removed from the oven.

## Statistical analyses

All analyses were conducted in the statistical programming environment R version R−4.1.1 (*R Core Team, 2021*). Elevational variations in eleven traits of *S. carpatica* (Table 2) were analysed for all 383 plants from 40 elevational sites with the linear mixed-effect models (*Field, Miles & Field, 2013*) using the *lmer()* function in the *lme4* package (*Bates et al., 2015*). In these analyses, elevation was used as a continuous fixed effect and sample site was set as a random model component. Because preliminary data analysis revealed that above 2,000 m a.s.l. there were considerable rapid changes in values of some traits that could
**Table 3  Summaries of linear mixed-effects models for fitting *Soldanella carpatica* traits to elevation.** Elevational model range - $mod_{2370}$: 890–2,370 m a.s.l., $mod_{1980}$: 890–1,980 m a.s.l.; $P_{model\ selection}$–test between straight line and curvilinear line models; t, P-test of model significance at the significance level of 0.05.

| | Elevational model range | $P_{model\ selection}$ | Intercept a | Slope $b_1$ | Slope $b_2$ | t | P |
|---|---|---|---|---|---|---|---|
| Aboveground plant mass (g) | $mod_{2370}$ | 0.0048 | $7.340 \times 10^{-2}$ | $3.389 \times 10^{-4}$ | $-1.447 \times 10^{-7}$ | $-2.858$ | 0.0067 |
| | $mod_{1980}$ | 0.4402 | $3.368 \times 10^{-1}$ | $-6.000 \times 10^{--5}$ | – | $-2.099$ | 0.0436 |
| Root mass (g) | $mod_{2370}$ | 0.0212 | $3.236 \times 10^{-3}$ | $2.435 \times 10^{-4}$ | $-8.918 \times 10^{-8}$ | $-2.299$ | 0.0270 |
| | $mod_{1980}$ | 0.9100 | $1.579 \times 10^{-1}$ | $3.729 \times 10^{-6}$ | – | 0.180 | 0.8583 |
| Total leaf mass (g) | $mod_{2370}$ | 0.0217 | $9.614 \times 10^{-2}$ | $2.156 \times 10^{-4}$ | $-9.961 \times 10^{-8}$ | $-2.288$ | 0.0275 |
| | $mod_{1980}$ | 0.4140 | $2.800 \times 10^{-1}$ | $-6.098 \times 10^{-5}$ | – | $-2.444$ | 0.0201 |
| Scape mass (mg) | $mod_{2370}$ | 0.0005 | $-1.684 \times 10^{+1}$ | $8.759 \times 10^{-2}$ | $-3.207 \times 10^{-5}$ | $-3.587$ | 0.0009 |
| | $mod_{1980}$ | 0.9431 | $3.934 \times 10^{+1}$ | $8.811 \times 10^{-4}$ | – | 0.1860 | 0.8536 |
| Total flower mass (mg) | $mod_{2370}$ | <0.0000 | $-6.663 \times 10^{0}$ | $3.683 \times 10^{-2}$ | $-1.337 \times 10^{-5}$ | $-5.656$ | <0.0001 |
| | $mod_{1980}$ | 0.2335 | $1.747 \times 10^{+1}$ | $1.322 \times 10^{-4}$ | – | 0.112 | 0.9112 |
| Individual flower mass (mg) | $mod_{2370}$ | <0.0000 | $1.323 \times 10^{0}$ | $6.539 \times 10^{-3}$ | $-2.462 \times 10^{-6}$ | $-4.353$ | 0.0001 |
| | $mod_{1980}$ | 0.2902 | $5.780 \times 10^{0}$ | $-2.286 \times 10^{-4}$ | – | $-0.793$ | 0.4336 |
| Scape height (mm) | $mod_{2370}$ | 0.0833 | $2.012 \times 10^{+2}$ | $-4.333 \times 10^{-2}$ | – | $-5.472$ | <0.0001 |
| | $mod_{1980}$ | 0.8607 | $1.820 \times 10^{+2}$ | $-2.842 \times 10^{-2}$ | – | $-2.583$ | 0.0144 |
| Leaf blade diameter (mm) | $mod_{2370}$ | 0.0041 | $1.821 \times 10^{+1}$ | $1.262 \times 10^{-2}$ | $-6.600 \times 10^{-6}$ | $-2.912$ | 0.0059 |
| | $mod_{1980}$ | 0.2017 | $3.061 \times 10^{+1}$ | $-5.873 \times 10^{-3}$ | – | $-4.435$ | 0.0001 |
| Root:Shoot ratio | $mod_{2370}$ | 0.1649 | $3.407 \times 10^{-1}$ | $2.643 \times 10^{-4}$ | – | 5.000 | <0.0001 |
| | $mod_{1980}$ | 0.7610 | $4.210 \times 10^{-1}$ | $2.020 \times 10^{-4}$ | – | 3.088 | 0.0022 |
| Leaf mass fraction | $mod_{2370}$ | 0.8786 | $5.733 \times 10^{-1}$ | $-8.149 \times 10^{-5}$ | – | $-5.878$ | <0.0000 |
| | $mod_{1980}$ | 0.4429 | $5.791 \times 10^{-1}$ | $-8.600 \times 10^{-5}$ | – | $-4.463$ | 0.0001 |
| Flower aboveground mass fraction | $mod_{2370}$ | 0.6018 | $5.234 \times 10^{-2}$ | $1.832 \times 10^{-5}$ | – | 3.590 | 0.0009 |
| | $mod_{1980}$ | 0.5847 | $4.715 \times 10^{-2}$ | $2.234 \times 10^{-5}$ | – | 3.216 | 0.0029 |

heavily influence the overall elevational variations, analyses were run for two elevation ranges independently; the first model for each trait was constructed for the full investigated elevation gradient 890–2,370 m a.s.l. ($mod_{2370}$), whereas the second model, constructed for elevation gradient 890–1,980 m a.s.l. ($mod_{1980}$), did not contain plants from the highest sites above 2,000 m a.s.l. To account for non-straight-line responses of traits to elevation two mixed-effect models were constructed and evaluated for the both examined elevation ranges. The first model included elevation as a linear fixed effects ($Y = a + b_1 \times$ (elevation), where *a* denotes an intercept and $b_1$ regression coefficient) while the second model included elevation and elevation with quadratic term ($Y = a + b_1 \times$ (elevation) + $b_2 \times$ (elevation)$^2$, where *a* denotes an intercept and $b_1$ and $b_2$ denote regression coefficients). Comparison of these models allowed to determine whether plant traits had a linear or nonlinear relationship with elevation. Fits of these two models to the data were evaluated based on a likelihood ratio test and *Chi-Squared* statistic where a significant $P_{model\ selection}$ value at 0.05 significance level (Table 3) indicated significant improvement in the straight-line model upon addition of the quadratic term for elevation (*Dalgaard, 2008*). Comparisons of model fit were carried out using the *anova()* function from the base R installation.

## RESULTS

### Elevational variation in morphological traits

Models of traits elevational variations fitted to the 890–1,980 m a.s.l. and the 890–2,370 m a.s.l. ranges (Table 3) differed considerably between these both ranges for many morphological traits indicating significant change in *S. carpatica* morphology occurring at the highest sites located at elevations 2,085–2,370 m a.s.l (Fig. 3). In the 890–1,980 m a.s.l. range aboveground plant mass ($P_{1980}$ = 0.0436, Fig. 3A), total leaf mass ($P_{1980}$ = 0.0201, Fig. 3C), scape height ($P_{1980}$ = 0.0144, Fig. 3G) and leaf blade diameter ($P_{1980}$ = 0.0001, Fig. 3H) decreased with an increase in elevation, whereas root mass ($P_{1980}$ = 0.8583, Fig. 3B), scape mass ($P_{1980}$ = 0.8536, Fig. 3D), total flower mass ($P_{1980}$ = 0.9112, Fig. 3E) and individual flower mass ($P_{1980}$ = 0.4336, Fig. 3F) did not vary across elevations. However, in the 890–2,370 m a.s.l. range all the traits had significant relationship with elevation (Table 3). Aboveground plant mass ($P_{2370}$ = 0.0067, Fig. 3A), root mass ($P_{2370}$ = 0.0270, Fig. 3B), total leaf mass ($P_{2370}$ = 0.0275, Fig. 3C), scape mass ($P_{2370}$ = 0.0009, Fig. 3D), total flower mass ($P_{2370}$ < 0.0001, Fig. 3E) and individual flower mass ($P_{2370}$ = 0.0001, Fig. 3F) did not have elevational trend or slightly decreased from 890 m to ca. 2,000 m a.s.l., whereas above 2,000 m a.s.l. values of these traits decreased very considerably. Scape height ($P_{2370}$ < 0.0001, Fig. 3G) decreased in a straight-line manner with an increase in elevation in the 890–2,370 m range, whereas leaf blade diameter ($P_{2370}$ = 0.0059, Fig. 3H) decreased slightly from 890 m to ca. 2,000 m a.s.l., and above 2,000 m a.s.l. values of this trait were reduced more considerably.

Generally, all traits reduced considerably their values with an increase in elevation from 890 m a.s.l. to 2,370 m a.s.l. (Table 4). In this range aboveground plant mass decreased with an increase in elevation by 77.5%, total leaf mass decreased by 78.9%, scape mass decreased by 73.6%, total flower mass decreased by 68.5%, individual flower mass decreased by 46.3%, scape height decreased by 37.2% and leaf blade diameter decreased by 56.4% (Table 4).

The reductions in size of plant organs occurred almost merely at the highest elevations above 2,000 m a.s.l., at sites located at elevations 2,085–2370 m a.s.l., which constitutes ca. 20% uppermost portion of the investigated elevation gradient. From 890 m to 1,980 m a.s.l. total leaf mass decreased by 2.7%, leaf blade diameter decreased by 2.3%, aboveground plant mass decreased by 2.1% and plant height decreased by 1.8% per every 100 m increase in elevation. Percent changes in other traits in this elevation range were small and models used to their calculations were statistically not significant (Table 4). Contrary, from 1,980 m to 2,370 m a.s.l. rates of changes in all traits were very considerable; aboveground plant mass decreased by 12.1%, root mass decreased by 11.3%, total leaf mass decreased by 11.2%, scape mass decreased by 16.2%, total flower mass decreased by 15.0%, individual flower mass decreased by 9.1%, scape height decreased by 3.6% and leaf blade diameter decreased by 7.2% per every 100 m increase in elevation (Table 4).

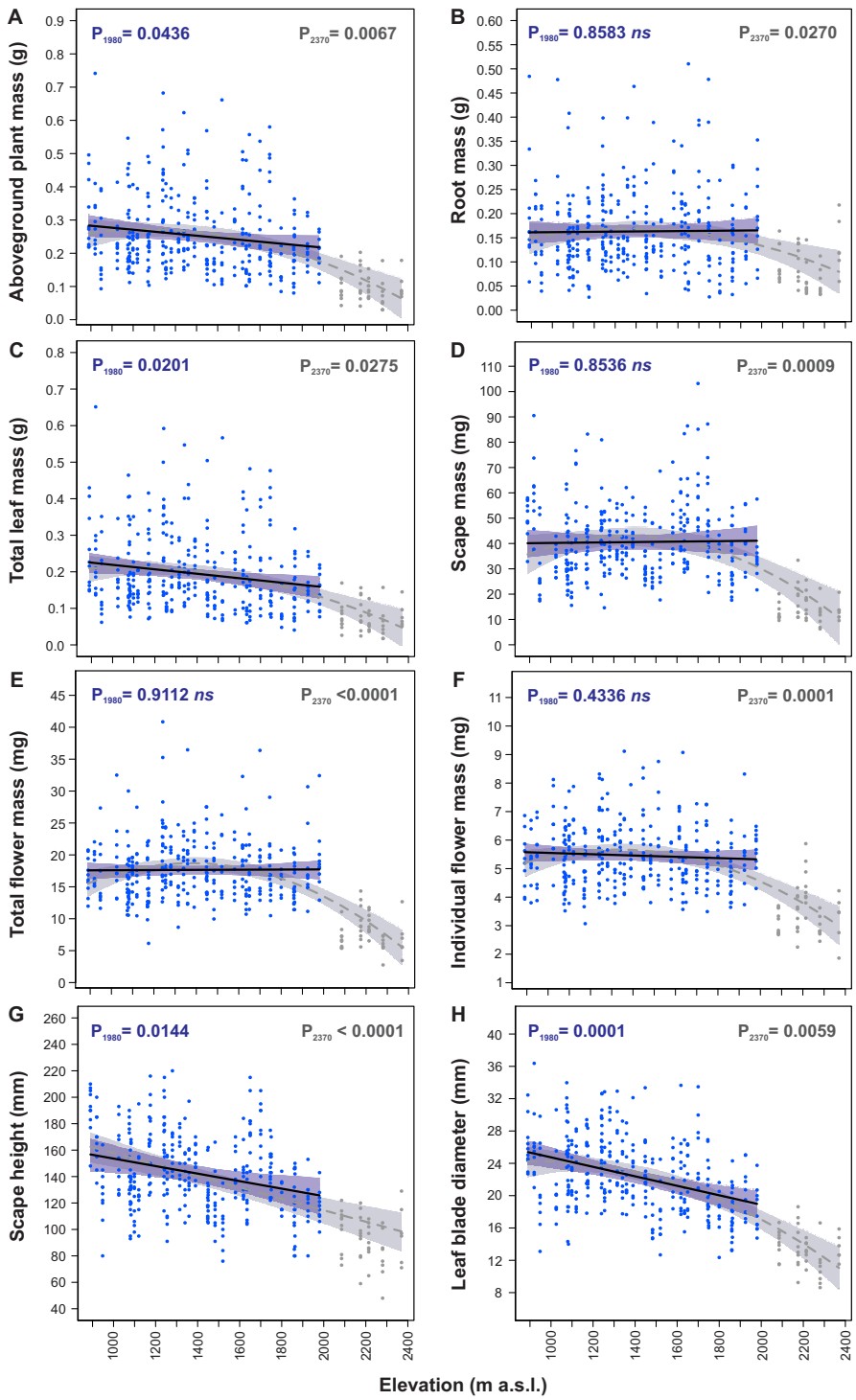

**Figure 3** **Elevational variation in morphological traits of *Soldanella carpatica*.** Solid lines represent the result of the mixed-effect models fitted to the 890–1,980 m a. s. l. range, interrupted lines represent models fitted to the entire 890–2,370 m a.s.l. range, the surrounding bands represent the 95% confidence intervals; $P_{1980}$, $P_{2370}$ –*p*-value of model significance fitted for 890–1,980 m a.s.l. and 890–2,370 m a.s.l. elevation ranges, respectively; ns –non significant model at 0.05 significance level.

### Elevational variation in dry mass allocation traits

Overall, *S. carpatica* plants allocated on average 39.8% of their total dry mass to root, 45.2% to leaves, 10.4% to scape, and 4.7% to flowers. Of the aboveground plant mass, 74.7% was allocated to leaves, 17.3% to scape and 8.0% to flowers.

All the dry mass partitioning traits were significantly fitted to elevation by straight-line models (Table 3, Fig. 4). Models constructed for both elevation ranges, the 890–1,980 m and the 890–2,370 m a.s.l. were almost identical for these traits. The root:shoot ratio increased with increasing elevation ($P_{1980} = 0.0022$, $P_{2370} < 0.0001$, Fig. 4A), whereas leaf mass fraction decreased with an increase in elevation ($P_{1980} = 0.0001$, $P_{2370} < 0.0001$, Fig. 4B). Moreover, flower aboveground mass fraction increased with an increase in elevation ($P_{1980} = 0.0029$, $P_{2370} = 0.0009$, Fig. 4C).

Within the entire elevation range from 890 m to 2,370 m a.s.l. the root:shoot ratio increased by 61.0%, leaf mass fraction decreased by 24.3%, and flower aboveground mass fraction increased by 42.8% (Table 4). The rates of these traits changes per 100 m increase in elevation had similar values in the 890–1,980 m a.s.l. and 1,980–2,370 m a.s.l. elevational ranges. In the 890–1,980 m a.s.l. range the root:shoot ratio increased by 3.4%, leaf mass fraction decreased by 1.7% and flower aboveground mass fraction increased by 3.3% per 100 m increase in elevation. Concurrently, in the 1,980–2,370 m a.s.l. range the root:shoot ratio increased by 5.3%, leaf mass fraction decreased by 1.5% and flower aboveground mass fraction increased by 2.2% per 100 m increase in elevation (Table 4).

## DISCUSSION

*Soldanella carpatica* varied across the elevation gradient adjusting its morphological and dry mass traits to environmental factors correlated with elevation. Along most part of the studied elevation gradient, from the base at ca. 900 m a.s.l. to ca. 2,000 m a.s.l., total dry mass of photosynthetic tissue, leaf size and scape height decreased steadily in a clinal manner with an increase in elevation, whereas dry mass investment in roots and flowers as well as individual flower mass did not vary with elevation. However, at the highest elevations considerable changes occurred in most of the traits leading to abrupt plant miniaturization. Accordingly, plants growing at elevations above 2,000 m a.s.l. were distinctly smaller in size, had lower aboveground dry mass, reduced leaf size and mass, reduced root, scape, total as well as individual flower mass, as compared to plants from sites at elevations below 2,000 m a.s.l.

### Elevational variation in aboveground vegetative organs

Maintaining relatively unchanged phenotype by *S. carpatica* across a wide 1,100-m elevation range from ca. 900 m to 2,000 m a.s.l. that corresponds approximately to ca. 6 K decrease in mean annual temperature with an increase in elevation suggests that this species is particularly well adapted to live across wide elevation gradients. Such specialised adaptations were found recently in the alpine species *Soldanella pusilla* that grows in mountains of the European Alpine System. *S. pusilla* is a species extremely adapted to tissue formation and growth under conditions of very low temperatures (*Körner et al., 2019*). That species resumes growth in mid-winter being at the time covered by a 2–3 m thick snowpack

**Table 4 Fitted values of *Soldanella carpatica* traits and percentage traits changes across elevations.** All percentage changes in trait values are referred to fitted values at the base of elevational gradient at 890 m a.s.l. Trait values at 890 m a.s.l. were fitted by linear mixed-effects models constructed for the 890–1,980 m a.s.l. elevational ranges, values at 2,370 m a.s.l. were fitted by models constructed for the 890–2,370 m a.s.l. elevational ranges, and values at 1,980 m a.s.l. were averaged based on the fitted values from the both models; ns, not significant model used for fitted values estimation.

|  | Fitted value at 890 m a.s.l. | Fitted value at 1,980 m a.s.l. | Fitted value at 2,370 m a.s.l. | Change per 100 m increase in elevation in the 890–1,980 m a.s.l. range (%) | Change per 100 m increase in elevation in the 1,980–2,370 m a.s.l. range (%) | Total change with an increase in elevation from 890 m to 2,370 m a.s.l. (%) |
|---|---|---|---|---|---|---|
| Aboveground plant mass (g) | 0.2834 | 0.1976 | 0.0639 | −2.1 | −12.1 | −77.5 |
| Root mass (g) | 0.1612 | 0.1505 | 0.0794 | + 0.2 *ns* | −11.3 | −50.7 |
| Total leaf mass (g) | 0.2257 | 0.1459 | 0.0476 | −2.7 | −11.2 | −78.9 |
| Scape mass (mg) | 40.12 | 35.97 | 10.60 | + 0.2 *ns* | −16.2 | −73.6 |
| Total flower mass (mg) | 17.58 | 15.79 | 5.54 | + 0.1 *ns* | −15.0 | −68.5 |
| Individual flower mass (mg) | 5.58 | 4.97 | 2.99 | −0.4 *ns* | −9.1 | −46.3 |
| Scape height (mm) | 156.7 | 120.6 | 98.5 | −1.8 | −3.6 | −37.2 |
| Leaf blade diameter (mm) | 25.4 | 18.2 | 11.1 | −2.3 | −7.2 | −56.4 |
| Root: Shoot ratio | 0.6008 | 0.8425 | 0.9671 | + 3.4 | + 5.3 | + 61.0 |
| Leaf mass fraction | 0.5025 | 0.4104 | 0.3802 | −1.7 | −1.5 | −24.3 |
| Flower aboveground mass fraction | 0.0670 | 0.0900 | 0.0958 | + 3.3 | + 2.2 | + 42.8 |

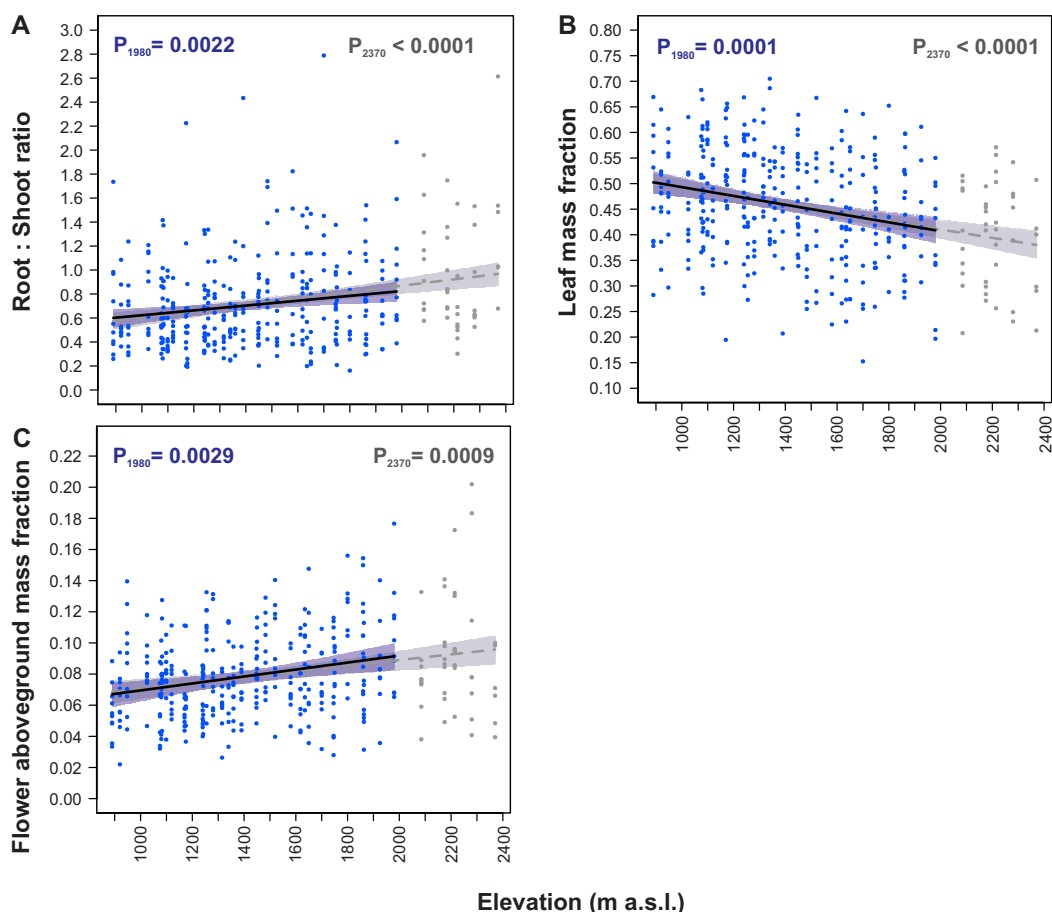

**Figure 4** **Elevational variation in *Soldanella carpatica* dry mass allocation traits.** Solid lines represent the result of the mixed-effect models fitted to the 890–1,980 m a.s.l. range, interrupted lines represent models fitted to the entire 890–2,370 m a.s.l. range, the surrounding bands represent the 95% confidence intervals; $P_{1980}$, $P_{2370}$ –$p$-value of model significance fitted for 890–1,980 m a.s.l. and 890–2,370 m a.s.l. elevation ranges, respectively.

and develops its entire flowering shoot under snow at 0 °C. Moreover, *S. pusilla* has an enormous carbohydrate reserves, mainly stored in below ground tissue, that would support basic metabolism for at least two entire years under snow (*Körner et al., 2019*). Although no similar studies are available for *S. carpatica*, the growth of this species may also not be significantly limited by low temperatures across the studied elevation gradient. Consequently, reductions in leaf size and total leaf mass in *S. carpatica* as observed from low to high elevations can represent rather adaptation to environmental conditions that change across elevations rather than restriction in growth caused by harsh environment of high elevations. Accordingly, reduction of mountain herbs size with an increase in elevation is a well-known elevational trend reported for plants representing different families, for example, Asteraceae (*Takahashi & Matsuki, 2017*; *Kiełtyk, 2018*; *Kiełtyk, 2021a*; *Kiełtyk, 2021b*; *Rathee et al., 2021*), Campanulaceae (*Maad, Armbruster & Fenster, 2013*), Fabaceae (*Cruz-Maldonado et al., 2021*), Gentianaceae (*He et al., 2017*), Poaceae (*Cruz-Maldonado*

*et al., 2021*), Zingiberaceae (*Paudel et al., 2019*). The general trend of plant size reduction toward high elevations contrasts, however, with the high metabolic capacity of plant tissues, especially leaves, at high elevations (*Körner, 2021*). Therefore, the fact that root, stem and flowers of *S. carpatica* were unresponsive to the 1,100-m elevation gradient from ca. 900 m to 2,000 m a.s.l. despite continuous reduction in dry mass of photosynthetically active tissue could be explained by increased photosynthetic leaf tissue efficiency that guarantees adequate assimilates supply for producing and maintaining unchanged rest of plant organs in harsher climatic conditions associated with increasing elevation. It could be therefore assumed, that because reduced photosynthetic organ size and mass at higher elevations fully satisfy plant demand for products of photosynthesis and plant produces and stores enough assimilates for growth, reproduction and year-to-year survival due to high metabolic efficiency of leaf tissue, there is no advantage in allocation in high elevations as much of dry mass to leaves as in low elevations, where inter-specific competition for light is stronger (*Weppler & Stöcklin, 2005*; *Callaway et al., 2002*; *Read et al., 2014*) and where increased investment in photosynthetically active tissue may be essential for capture and utilization of sufficient light quantity. Moreover, the reduction in plant aboveground size with an increase in elevation, that in *S. carpatica* is caused by reduction in leaf size and mass, can be advantageous adaptation to decreased nutrients availability in soils (*Sveinbjörnsson et al., 1995*) and climatic constraints at high elevations because smaller plants have lower resource requirements, as proposed by the 'resource-cost compromise' hypothesis (*Herrera, 2005*; *Zhao & Wang, 2015*). Additionally, the smaller are plants with smaller leaf rosettes growing close to the ground the more they decouple their climate from the ambient due to reduction in surface area that is exposed to cold air (*Körner, 2021*). Being closely attached to the ground also increases plant resistance to environmental factors such as strong winds as well as to freezing events due to heat accumulation by the leaf canopy close the ground (*Fabbro & Körner, 2004*; *Cruz-Maldonado et al., 2021*). Therefore, decreasing plant aboveground size with an increase in elevation can represent adaptive adjustment allowing local populations to efficiently use limited resources under increasingly less favourable climatic conditions associated with increasing elevation.

### Elevational variation in reproductive organs

Dry mass investment in flowers as well as flower size expressed by mean flower mass were not affected by elevation along a 1,100-m elevation gradient ranging from ca. 900 m a.s.l. to 2,000 m a.s.l. Despite increasingly more growth limiting conditions due to reduced temperature and resource limitation with an increase in elevation, *S. carpatica* retained steady dry mass investment in flowers across the wide elevation range. Similar pattern of elevational variation in flower mass was found in *Bellidiastum michelii*, where flower head mass, total flower mass as well as individual flower mass did not change considerably along ca. 900-m elevation gradient and decreases in these traits were visible only at the highest elevations of the species range (*Kiełtyk, 2021a*). Furthermore, unchanged floral mass investment across the wide 950-m elevation gradient was found in total mass of *Senecio subalpinus* flower heads, however, at the level of an individual flower head both total tubular flower mass as well as total ligulate flower mass increased toward high elevations (*Kiełtyk,*

*2021b*). Maintaining unchanged total mass of *S. carpatica* flowers along with reduction in total leaf mass with an increase in elevation across ca. 1,100-m elevation gradient resulted in the increase in flower aboveground mass fraction. Meanwhile, the increased portion of dry mass allocated to flowers may suggest that reproductive investment is increasingly important strategy for local *S. carpatica* populations persistence with an increase in elevation.

The plant height, a trait being related with species competitiveness for light in species with leafy stems, that is often used as a proxy for plant size, was commonly found to decrease with an increase in elevation in studies on intraspecific plant variations (*e.g.*, *He et al., 2017*; *Paudel et al., 2019*; *Kiełtyk, 2021b*). However, in case of *S. carpatica* plant height as expressed by the scape height is not related to plant size nor competitive ability for light because the species has all its leaves gathered in basal rosette and the only function of scape is to give support for flowers and eventually fruits clustered at the top of scape. Thus, height of scape in *S. carpatica* defines elevation on which flowers are positioned above the ground, and functionally this trait is related to reproductive process by exposing flowers above the ground to attract pollinators and support fruits when dispersing seeds. With an increase in elevation from ca. 900 m to 2,000 m a.s.l. height of *S. carpatica* scape decreased steadily, whereas amount of dry mass allocated to scape remained unchanged, what resulted in producing thicker and tougher scapes with flowers elevated lower above the ground toward higher elevations. Because height of leaf canopy of surrounding vegetation generally also decrease with an increase in elevation (*Leuschner & Ellenberg, 2017*) reduced scape height prevents flowers to protrude too high into well-stirred cold air above the calm aerodynamic boundary layer created by leaf canopy (*Dietrich & Körner, 2014*). It has been shown in the alpine environment that the closer the ground the higher air temperatures are whereas wind speed rises logarithmically with distance from the ground (*Körner, 2021*). Therefore, reducing the scape height toward higher elevations follows the general vegetation trend of reduced foliage layer height and can enhance decoupling flowers from the colder free atmosphere. Meanwhile, it has been shown that warmer flowers can more efficiently attract pollinators as pollinators besides flower colour are also sensitive on flower temperature (*Creux et al., 2021*), in colder climates preferring warmer flowers (*Norgate et al., 2010*). Additionally, warmer flowers have thermal benefits during maturation of sexual organs, pollen germination and pollen tube growth (*Dietrich & Körner, 2014*) and it has been found that experimentally increased temperature and sheltering from wind of *Ranunculus acris* plants growing on mountains resulted in increased seed production (*Totland & Eide, 1999*). Thus, decreasing scape height in *S. carpatica* with an increase in elevation can be a selective advantage of developing flowers in milder micro-climate closer to the ground as maintaining higher floral temperatures in cool alpine environment is often critical for the successful reproduction of high-mountain plants (*Dietrich & Körner, 2014*).

## Elevational variation in root mass

Elevational variation in root mass of *S. carpatica* was analogous to that in flower mass as there was no significant change in root mass in a wide range of elevations from ca. 900 m to 2,000 m a.s.l. whereas above 2,000 m a.s.l. this trait decreased significantly. Steady

root mass investment across ca. 1,100-m elevation range accompanied by decrease in total leaf mass with increasing elevation resulted in increasing the root:shoot ratio from low to high elevations. Because plants can adjust their morphological traits such as root length, density, branching as well as proportion of fine to coarse roots (*Weemstra et al., 2021*) to micro-mosaics of belowground conditions such as water and nutrient availability or soil rockiness, the increased *S. carpatica* belowground mass fraction toward high elevations may indicate an increased importance of resource acquisition and/or assimilates storage in resource limited and climatically more growth restricting and unpredictable environmental conditions of high elevations. Moreover, the decreased availability of nutrients may result not only from decreasing soil nutrients content toward high elevations (*Körner, 2021*) but also from increased belowground competition between plants for limited resources. It has been suggested that in abiotically severe environments of high elevations competition strength decreases as a result of reduced number of species that withstand unfavourable conditions as well as due to restrictions in resource acquisition by plants caused by severe physical conditions (*Callaway, 1998*). This relation may characterize aboveground plant competition for space and light as height of leaf canopy decrease and vegetation becomes less dense with increasing elevation (*Nagy & Grabherr, 2009*). However, because soil nutrients content and soil depth generally decrease with an increase in elevation (*Sveinbjörnsson et al., 1995*; *Körner, 2021*) nutrients availability for plants also decreases what can result in increasing belowground resource competition. Thus, dry mass investment in roots can be increasingly important for resource foraging with an increase in elevation as evidenced by unreduced *S. carpatica* root mass from ca. 900 m to 2,000 m a.s.l. despite accompanying continuous reduction in aboveground plant mass. Furthermore, increasing the root:shoot ratio with an increase in elevation supports the hypothesis that in colder and harsher environments plants tend to allocate greater portion of their mass in belowground organs (*Bloom, Chapin III & Mooney, 1985*; *McConnaughay & Coleman, 1999*). Increased root mass fraction may be advantageous adjustment in *S. carpatica* to acquire and/or store enough resources for growth, survival and completing its life history cycle in high elevation environments.

## Plant miniaturization at the highest elevations

The most pronounced pattern of elevational variation in *S. carpatica* is abrupt plant miniaturization on sites located in the 20% uppermost part of the studied elevation gradient. Rapid reduction in overall plant size may suggests that this species encounters considerably more growth limiting conditions above 2,000 m a.s.l. as compared to lower elevations. The reduction in *S. carpatica* size cannot be explained merely by plant response, by means of phenotypic plasticity and/or genetic adaptation, to limiting climatic conditions of high elevations because climate factors change gradually with an increase in altitude (*Nagy & Grabherr, 2009*), as exemplified by mean annual temperature decrease by 0.55 K per every 100 m increase in elevation in mountains of temperate regions (*Körner, 2021*). Therefore, it could be other than climatic factors alone that trigger off morphological changes in *S. carpatica* at the highest elevations of the studied gradient. It is likely that the plant miniaturization above 2,000 m a.s.l. was related to changes in orographic conditions,

*i.e.,* increased steepness and rockiness of mountain slopes toward high mountain tops. The five highest sites on which size of *S. carpatica* was considerably reduced were located at elevations 2,085–2,370 m a.s.l. on steep rocks of Czarny Mięguszowiecki Wierch peak (2,405 m a.s.l.). This environment is characterized by harsh climatic conditions and lack of accumulated snow cover in winter due to frequent strong winds and high slopes inclination. *S. carpatica* growing on rocky initial soils at the highest elevations of the studied gradient was exposed to particularly unfavourable conditions caused by strong winds and lack of protective snowpack in winter, that was found to be important for development and growth of other species from *Soldanella* genus in the European Alps, namely *S. pusilla* (*Körner et al., 2019*). Dwarfism of *S. carpatica* at the highest elevations could be, therefore, an adaptive response to abrupt deterioration in growing conditions caused by simultaneous interactions of climate, soil and orographic conditions. Similar pattern of rapid change in plant size across elevations was found in high elevation taxon of the *Solidago virgaurea* L. complex, where plants growing at the highest elevations had distinctly reduced plant height and rosette diameter (*Takahashi & Matsuki, 2017*) as well as lower number of flower heads (*Sakurai & Takahashi, 2017*) as compared to plants from lower elevations. Furthermore, reduced size of *S. carpatica* at high elevations is in line with studies on local adaptation to high elevation environment in *Arabidopsis arenosa* (*Knotek et al., 2020*; *Wos et al., 2022*) that revealed that elevation, a sharp environmental gradient, impose consistent selective pressures on plant growth and reproduction, leading to emergence of striking differences in life-history strategies and growth forms (*Wos et al., 2022*). Furthermore, in *A. arenosa* parallel local adaptation to alpine environment occurred independently in several mountains from different low elevation lineages and resulted in emergence of genetically distinct alpine populations characterized by smaller plants (*Knotek et al., 2020*).

It is important to note that alpine zone of the European Alpine System was historically colonized by species of the *Soldanella* genus, namely *S. minima* and *S. pusilla*, species of the 'Tubiflores' group characterized by overall dwarfism and reduced biomass production (*Rurik et al., 2024*). Moreover, adaptation to alpine environment occurred also at intraspecific level in montane taxa of the 'Soldanella' group to which *S. carpatica* belongs (*Rurik et al., 2024*). Results of the present study on *S. carpatica* elevational variation generally support statement that alpine adaptation in populations of species from the 'Soldanella' group is characterized by overall dwarfism trend with plant size reductions, but with no significant reduction in flower size and morphology, and maintaining the typical 'Soldanella'-like appearance, as opposed to the snowbed specialists from the 'Tubiflores' group that have significantly reduced flower structures (*Rurik et al., 2024*). However, it should be noted that contrary to the predictions that overall dwarfism evolves when a niche shifts from a forest to an alpine zone (cf. *Rurik et al., 2024*) there were no abrupt changes in morphology between forest and alpine populations of *S. carpatica*. Instead, analysis of the species variation along the wide continuous elevation gradient enabled to reveal that reductions in *S. carpatica* size and biomass were gradual from 900 m to 2,000 m a.s.l., what may suggest that populations at given elevations are locally adapted to elevational gradient of environmental conditions and that the transition in environmental conditions between forest and alpine zones is not sharp for low stature species growing close to the ground.

Only at the highest elevations above 2,000 m a.s.l., in the upper part of the alpine zone and in the subnival zone where plants are not sheltered by dense vegetation layer, changes in *S. carpatica* morphology were very considerable, resulting in abrupt plant miniaturization, including also flower size and mass. This points that at these elevations important changes in the *S. carpatica* life conditions occur. Furthermore, recent analyses of treats evolution within the *Soldanella* genus showed that in Carpathian species of the 'Soldanella' group (*i.e.*, *S. carpatica*, *S. hungarica*, *S. major* and the *S. marmarossiensis* group) for survival in alpine zone niches, dwarfism alone, without reduction in the flower number and structure has proven to be sufficient (*Rurik et al., 2024*). The results of the present study on *S. carpatica* elevational variation support this statement, with the reservation that the observed reduction in floral investment at the highest elevations may be crucial for plant survival and persistence in hostile environment at the high end of the species vertical range.

It is noteworthy that environmental conditions that triggered off abrupt changes in morphology of alpine plants may also hinder predictions on plants upward migrations in mountains (*Frei, Bodin & Walther, 2010*; *Wipf et al., 2013*; *Steinbauer et al., 2018*) as well as plant phenotypic responses to climate warming if local orographic conditions or topographic microhabitat diversity are not considered (*Kulonen et al., 2018*). It is likely that increased rockiness and steepness of slopes and walls at high elevations of young mountains with glacial landforms shaped by geomorphic processes of physical weathering and denudation (*Kotarba, 1996*) coupled with lack of snowpack during dormant season, that causes freezing stress (*Körner, 2021*), increased frequency of strong winds, shallow initial soils and increased patchiness of fragmented vegetation (*Leuschner & Ellenberg, 2017*) may not constitute suitable habitats for plant species from lower elevations migrating upward mountain slopes despite the increase in mean annual temperature as predicted by future climate warming scenarios (*Engler et al., 2011*; *Mountain Research Initiative EDW Working Group, 2015*). Moreover, diversity of microhabitats in alpine landscapes creates mosaic of micro-environmental and life conditions, with range of thermal niches that can differ in mean temperature by as much as 8 K at the same elevation, and this exceeds around twice the worst climate warming scenario for the same region, as recently shown in the European Alps by *Körner & Hiltbrunner (2021)*. Furthermore, the difference in temperature by 8 K corresponds to ca. 1,450 m difference in elevation, what equals approximately to the differences in mean air temperature across the elevation range investigated in the present study. Such diversity of thermal conditions at given elevation may ensure availability of suitable conditions for cold adapted plants under warmed climate conditions. Therefore, when predicting plants phenotypic responses across elevations in mountains or possibility of upward plant species migrations caused by climate warming it is important to consider local dominant orographic as well as topographic conditions, *e.g.*, slope inclination, exposition, rockiness, terrain relief, because they can substantially modify the climate conditions experienced by plants at given elevation (*e.g.*, *Graae et al., 2018*).

This study revealed the pattern of elevational variation in *S. carpatica* caused by changes in environmental conditions correlated with elevation. It is plausible that observed changes in morphology of *S. carpatica* across elevations represent genomic based adaptation of

the species to inhabited niches, as was recently found, for example, in parallel adaptation to different mountain niches in model species *Arabidopsis arenosa* (*Knotek et al., 2020*; *Bohutínská et al., 2021*; *Wos et al., 2022*) and *Heliosperma pusillum* (*Szukala et al., 2023a*; *Szukala et al., 2023b*). In the context of the emergence of dwarf form of *S. carpatica* above elevation 2,000 m a.s.l. it is worth to note, that due to historically oscillating conditions radiated lineages from high mountain environments are capable of rapidly adopting to a wide range of environments, resulting in adaptive phenotypic evolution (*Pouchon et al., 2018*; *Rurik et al., 2024*). The emergence of such adaptations to high elevation environment in *Soldanella* genus could be additionally facilitated by frequent hybridization events and pervasive introgression among ancestral lineages of snowbells, that persisted throughout the entire evolutionary history of the genus (*Slovák et al., 2023*). Furthermore, these processes have the highest extent in the Carpathian species of the genus where they could greatly facilitate lineages diversification as well as enrichment of genetic pools of species (*Slovák et al., 2023*) enabling rapid adaptation to local environments. However, taking into account that in the Tatra Mts. two other *Soldanella* species also occur, namely *S. marmarossiensis* and *S. montana* (*Valachovič et al., 2019*), it cannot be precluded that some of the studied *S. carpatica* plants had hybrid origin as a result of gene flow from these species. However, it should be stressed that based on morphological characters all the studied individuals were determined as *S. carpatica*. Moreover, it cannot be completely ruled out that distinctly smaller phenotype of *S. carpatica* at the highest elevations may resulted from plastic, environmentally induced growth responses to limiting abiotic conditions, rather than represent genuine genetically based adaptation (ecotype). Furthermore, along with changes in climate factors across elevations, many other environmental factors, for example, geological substrate, moisture, soil properties and nutrient availability, slope exposure and inclination, and length of snow free period, could influence *S. carpatica* life conditions and the observed pattern of the species elevational variation. Therefore, to disentangle the contributions of plastic, environmentally induced responses, and of genetic differences among locally adapted populations, to morphological variation across elevation gradients studies involving common garden cultivation experiments and reciprocal plant transplantations to different elevations are required (*e.g.*, *Scheepens, Frei & Stöcklin, 2010*; *Hamann et al., 2016*; *Knotek et al., 2020*; *Wos et al., 2022*).

## CONCLUSIONS

Despite availability of numerous studies on elevational variations in plants relatively few of them are based on large number of elevational sites that enable modelling traits variation by continuous elevation variable and explore shape and trajectory of morphological changes across plants vertical ranges. Contrary to what may be expected, that dwarfism in alpine populations of mountain plants emerge when a niche shifts from a forest to an alpine zone, *S. carpatica* morphotype did not change significantly between these two contrasting zones. Instead, there were steady, gradual reductions in overall plant size and biomass from the mountain foothills at 900 m a.s.l. to the alpine zone at elevation 2,000 m a.s.l. that suggest adaptation of local populations to the gradient of climatic conditions across

elevations. However, a miniaturization of *S. carpatica* phenotype emerged on rocky slopes at the highest elevations in the upper part of alpine zone and in subnival zone. This rapid plant dwarfism may reflect adaptation to abrupt changes in orographic conditions *i.e.,* increased rockiness and steepness of mountain slopes that cause increase of harshness of abiotic conditions. Despite the plant miniaturization at the highest elevations, biomass partitioning traits changed gradually across the entire species elevation range. Thus, the abrupt changes in *S. carpatica* morphology on the highest elevation sites suggest that overall plant size and dry mass are strongly influenced by local environmental conditions to which they respond considerably by means of adaptation and/or phenotypic plasticity, whereas steady trajectories of dry mass partitioning changes across entire elevation gradient suggest that mass partitioning and adjustive changes in organs mass fractions in response to environmental conditions represent a stable evolutionary solution of the species. Moreover, morphological variations in *S. carpatica* conform to the observations that populations of species from the 'Soldanella' intrageneric group that are adapted to high elevation environments still maintain typical 'Soldanella'-like appearance, despite considerable reduction in overall size.

## ACKNOWLEDGEMENTS

The support of collecting the material in the field by Wiesława Kiełtyk and Anna Delimat is highly appreciated. I thank the reviewers and editor for their constructive comments that significantly enhanced the manuscript.

### Funding

This work was supported financially by the Cardinal Stefan Wyszyński University in Warsaw. The funders had no role in study design, data collection and analysis, decision to publish, or preparation of the manuscript.

### Grant Disclosures

The following grant information was disclosed by the author:
Cardinal Stefan Wyszyński University in Warsaw.

### Competing Interests

The author declares that they have no competing interests.

### Author Contributions

- Piotr Kiełtyk conceived and designed the experiments, performed the experiments, analyzed the data, prepared figures and/or tables, authored or reviewed drafts of the article, and approved the final draft.

### Field Study Permissions

The following information was supplied relating to field study approvals (i.e., approving body and any reference numbers):

Collection of plant material was approved by the Tatra National Park (Permission Bot/380, DBN.503.28/18).

## Data Availability

The raw measurements are available in the Supplementary File.

## Supplemental Information

Supplemental information for this article can be found online at http://dx.doi.org/10.7717/peerj.17500#supplemental-information.

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
