# Peer review of "Elevational variation in morphology and biomass allocation in carpathian snowbell Soldanella carpatica (Primulaceae)"

_PeerJ, doi:10.7717/peerj.17500_

## Round 0.1 · original submission · Major Revisions

The intraspecific variation exhibited by plants not only furnishes them with the capacity to thrive across diverse environments but also underscores their adaptive prowess. Every research endeavor aimed at unraveling the intricacies of environmental adaptations of plants holds immense value. I am sure that the findings of your study will serve as a crucial resource for future researchers. However, it is essential to address some technical aspects in refining your article. I recommend a thorough review of the reviewers' suggestions and a judicious consideration of each recommendation. If you find yourself in disagreement with any particular suggestion, it would be beneficial to provide clear and well-reasoned justifications for your perspective.

Reviewer 1 ·

Basic reporting

The paper appears to be well-written to the extent I can evaluate it, bearing in mind that English is not my native language. The literature background seems insufficient to me because my main concern is that the author overlooked most of the currently published studies focused on the model system. At least some of them are clearly linked to the major topic of the presented manuscript and, in my opinion, should be included in the introduction and the section about the studied species. These references should definitely be used to discuss the findings in the Discussion section. Overall, the structure of the article, figures, and tables seems to follow common standards and is relevant.

Experimental design

The experimental design appears correct and meaningful to me, but I have reservations about the selection of the model system, especially concerning the novel contributions it may bring compared to previously published studies focused on this topic.The methodology is generally well described, but in some places, it needs more details. Please refer to my general comments for specific suggestions.

Validity of the findings

Even though the findings are interesting, as mentioned several times, it is necessary to specify the rationale behind the selection of the model system. Additionally, hypotheses are missing in this study. The analytical part (statistical analyses) appears to be relevant and well-performed based on the information available to me. However, further clarification and details regarding the rationale and hypotheses would strengthen the overall scientific rigor and clarity of the study.

Additional comments

The manuscript, " Elevational variation in morphology and biomass allocation in carpathian snowbell Soldanella carpatica (Primulaceae) " authored by Kiełtyk 2024, focuses on the phenotypic variation of a plant with a large ecological amplitude in the Western Carpathians, using Soldanella carpatica as a model system. The author analyzed both above- and below-ground organs in hundreds of individuals from 40 localities. The crucial finding suggests a major shift in phenotype occurring at an altitude of around 2000 m a.s.l., where a significant shift towards miniaturization is observed.
The topic of species with a large ecological amplitude is inherently attractive and interesting for a broad scientific audience, making this article potentially appealing for readers of PeerJ Journal. Particularly commendable is that the study is performed by a single author, which is relatively rare nowadays. However, I have several concerns regarding this study that need clarification and improvement before considering it for publication in PeerJ. At its current stage, I cannot recommend it for acceptance in PeerJ.

My primary concern pertains to the overall scientific concept, or better to say its originality. While I acknowledge the appeal of investigating the phenotypic variation of plant species across altitudinal gradients, an area of interest not only for botanists but also for ecologists and evolutionary biologists, I find it noteworthy that the author has previously published several papers on different species from various families within the same geographic region and esentailly the same major objectives and questions (Kiełtyk 2018 – Alpine Botany, Kiełtyk 2021a – PeerJ, Kiełtyk 2021b – Alpine Botany). I question the rationale behind selecting this particular model species for the examination of phenotypic variation in the studied area. What sets this study apart in terms of novelty? It seems to replicate a similar question with a comparable design in the same geographic area, albeit with different species. What makes it compelling for readers, and what unique contributions does it offer to the existing body of literature beyond employing different model species? Clearly articulating the distinctive aspects and contributions of the present study in comparison to the author's prior work is not only essential for enhancing its overall significance and relevance but, from my perspective, is fundamental when considering the publication of this paper in a reputable journal such as PeerJ.
Secondly, I find it problematic that the author overlooked recently published literature about the studied system. In my opinion, this literature is not only essential for introducing the model system comprehensively with important background knowledge but also crucial for the interpretation and discussion of the findings in the Discussion section. (Please refer to the specific comment below for more details.)

Another points
Abstract
I would not say that S carpatica is ‘alpine plant’. It does not grow neither in Alps nor exclusively in alpine vegetation belt. It is mountain species. Please improve this.
Studied species
I am surprised that the author decided not to cite current studies focused on the genus Soldanella, which are relevant to the studied topic. Specifically, the detailed distribution range, including ecology and sociological associations, is thoroughly investigated in Štubňová et al. 2019. This study clearly illustrates the ecological range of the species. Furthermore, two later studies, Slovák et al. 2023 (Systematic Biology) and Rúrik et al. 2024 (The Plant Journal), not only include phylogenomics analyses, incorporating S. carpatica and revealing its phylogenetic position and evolutionary relationships, but also encompass numerous populations from various altitudes and comments on this phenomenon. In Rúrik et al. 2024, the authors reconstructed trait evolution, and within the analyses of traits, ecology (alpine versus mountain versus large ecological amplitude) was also considered. I assume that this investigation is more than relevant to the presented study. Why did the author not include these studies in the Introduction, Material and methods - Studied species, but especially in the Discussion section? I would definitely expect more details about this species in introductory parts.

Field sampling and traits measurements

The author sampled S. carpatica along gradients in the Polish part of the Tatra Mountains. However, in this region of the Tatra Mountains, S. marmarossiensis also co-occurs with S. carpatica, and at lower elevations, it might grow in close proximity to S. montana. Current molecular analyses indicate extensive hybridization between most Carpathian species of Soldanella. Did the author consider potential hybridization with related species? It is crucial to address this potential issue, as the inclusion of hybrids might significantly impact the results of the analyses. This aspect should be explicitly mentioned and discussed in the text, as it could have implications for the interpretation of the study's findings.
lines 184-192 The author describes elevation zones in the Tatra Mountains. However, at this point, it seems somewhat out of context and may not offer significant information to readers. I would recommend suggesting to the author to place this information within the context of S. carpatica ecology and distribution. This way, it can be more seamlessly integrated into the narrative, providing readers with a clearer understanding of how elevation zones relate to the distribution of S. carpatica in the Tatra Mountains.at lines 206-207 the author mentioned the method used to measure stem height: 'Scape height was assessed as a distance between the plant base just below rosette leaves and the top of inflorescence; during the measurement, the scape was straightened.'
The text does not provide information on the total number of plants collected and analyzed, nor does it specify if all traits were measurable in every plant. It is crucial to include such details to provide a comprehensive understanding of the scope and completeness of the study. Adding this information will contribute to the transparency and thoroughness of the research.

In the chapter "Field Sampling and Traits Measurements," it would be beneficial to provide a list of all the measured traits or, at the very least, make a reference to a table where readers can find this information. Although the manuscript mentions that the details can be found in Table 2, it would enhance the clarity and accessibility of the information if there was a specific mention within this chapter about where readers can refer to for a comprehensive list of measured traits. This ensures that readers are directed to the relevant table for a more detailed understanding of the traits under consideration.
Discussion
In lines 317 and 318, the author stated that S. carpatica is closely related to the Alps-dwelling S. pusilla. Recent genomic studies, however, reveal that these two species are not as closely related as implied by the study of Zhang et al. 2001 (American Journal of Botany), which is unfortunately also not cited in this manuscript. The closest relatives are S. minima and S. alpina (refer to Slovak et al. 2023), and they are preferably snow bed species. As previously mentioned, the ecological characteristics of Soldanella species, especially with regard to their large ecological amplitude and adaptation to forest versus alpine vegetation belts, are extensively discussed in Štubňová et al. 2019 and particularly in the recently published study by Rúrik et al. 2024. Last but not least, S. pusilla is not exclusively growing in the European Alps. It also occurs in the Bulgarian Mountains and Southern Carpathians. If the author meant the European alpine system, this is different from the European Alps. I would recommend that the author will improove his knowledge about studied system from mentioned literature and discusses his findings in light of the information provided in these studies.

In the last paragraph about miniaturization, it would be highly beneficial to discuss interesting cases of miniaturization and adaptation to the alpine zone in the genus Arabidopsis (see Kolař et al., 2016 - Molecular Ecology, Knotek et al. 2021 - Frontiers in Plant Sciences, Bohutínska et al. 2021 – PNAS, Šrámková et al. 2019 - Plant Systematics and Evolution, etc.). There are interesting genomic studies indicating parallel evolution of these alpine forms, which indeed also resulted in speciation events, and comparing it with the studied system and other species studied would enhance the depth of the discussion.

Reviewer 2 ·

Basic reporting

The manuscript has a typical, correct structure of a scientific publication. The style is clear and understandable, although I do not feel competent to assess the correctness of the language. The exhaustive bibliography includes over 100 items, properly cited. The number of tables and figures is correct, properly cited. Raw data is available and correct. The introduction provides an appropriate introduction and background to the research. Research and results related to hypotheses.

Experimental design

The manuscript describes own original research of the author, the scope of which is consistent with the scope of the journal. Modern methods are properly and sufficiently described. The research was carried out according to the given methods. The author obtained the required permits to collect material in a protected area.
The research questions are very simple and not original. The author undertook to investigate changes in the morphology and biomass of the species Soldanella carpatica with increasing altitude. The author himself has already published the results of similar research, for other species, including in PeerJ (DOI:10.7717/peerj.11286), but also in other journals (DOI:10.7717/peerj.11286 ; DOI: 10.1007/s00035-017-0197-7). All of them were carried out in the Western Carpathians, so the spatial scope is regional. The research included in this manuscript concerns a species endemic to the Western Carpathians. Repeating studies for different taxa may be justified, but such justification is missing in the study.
It would be better if the author pointed out his previous findings and commented on current research comparing it with previous research.

Validity of the findings

All necessary data has been provided and is correct. The conclusions are not entirely related to the hypotheses and are rather part of the discussion.
The work replicates previous research, and the benefits of such replication have not been justified.
The revised work should contain analyzes comparing previously studied species and aim at general conclusions. New goals and assumptions should be set.

Additional comments

no comment

Reviewer 3 ·

Basic reporting

An interesting and valuable study with several relevant results. A well chosen model species, occurring in a very wide range of altitudes and vegetation types in the area of interest. Extensive and detailed review of the theoretical issues of variation in plant size as a function of altitude and related plant adaptation strategies. But, insufficient baseline characterisation of the model species Soldanella carpatica, not taking into account the results of more recent taxonomic work on the whole genus Soldanella, including several recent studies (e.g. the latest one, Rurik et al., The Plant Journal, 2024).

Experimental design

Clearly stated questions and initial hypotheses. But, insufficient characterization of the study area (lines 179-192). Information on geological substrate, soils, basic ecological characteristics of study sites (e.g., data on exposure to cardinal directions, slope gradient, soil properties, etc.) is lacking. In addition to altitude, the wide range of substrate characteristics can largely influence the variability in plant size and the amount of biomass produced. In this context, it is necessary to define the specific abiotic conditions related to all the study sites. Alternatively, if all study sites had the same abiotic conditions (geological substrate, soils, amount of available nutrients, etc.) and differed only in altitude, this should be explicitly stated. The Methods chapter lacks an explanation under which conditions the trait measurements were made (e.g. whether they were living or dried plant material used). This information only appears in the Table 2.

Validity of the findings

The results obtained, their evaluation and interpretation, as well as the discussion, are limited to an assessment of the dependence on altitude only. Other possible influences (e.g. several other abiotic conditions at the study sites mentioned above) are not considered at all (see above comment to experimental design). This can be considered the weakest aspect of the manuscript under review. In the Discussion section (lines 331-333), it is requested to specify with which systematic groups the author compares the results obtained. Indeed, it is clear that the trends considered may be quite different in different systematic groups. Ideally, a comparison in the genus Soldanella or in the family Primulaceae would be desirable. The discussion could have been supplemented by the results of the recent study by Rurik et al. 2024 mentioned above.

Additional comments

The results (line 244) give a range of elevations of 890-1980 m. In contrast, the Discussion (lines 302-311) gives values of 890-2000 m. I see no reason why they are not used uniformly. The use of any reference in the Conclusions chapter (line 522) seems strange.

---

## Round 0.2 · accepted · Accept

I would like to thank you for accepting the referees' suggestions and improving your article based on their input. I believe your manuscript is now ready for publication. We look forward to your next article.

Reviewer 1 ·

Basic reporting

This is the second revision of this manuscript that I'm reviewing. Despite several critical points raised in the first review, I must acknowledge that the author has done an excellent job and positively addressed all of my concerns, significantly enhancing the manuscript's quality. Specifically, the author paid special attention to explaining the originality of their study and highlighting their findings. Additionally, the second major shortcoming, namely the lack of relevant literature regarding the studied system, has been effectively addressed. Therefore, in the current stage, I consider the manuscript suitable for acceptance in PeerJ. Nonetheless, minor changes still should be made.

Experimental design

Is clear and well explained.

Validity of the findings

The findings are of interest to a wide range of plant biology and ecology-focused scientific audiences and merit publication in PeerJ.

Additional comments

As already mentioned, the manuscript has been significantly improved and is suitable for publication. However, the introduction chapter has now extended to almost 5 pages, which I believe is too lengthy. Therefore, I would recommend that the author streamline and condense the information presented in the Introduction chapter while ensuring that none of the important points, including those added after revision, are excluded.

Reviewer 2 ·

Basic reporting

The manuscript is clear and self-contained. The corrections made significantly improved it.

Experimental design

The manuscript describes own original research of the author, the scope of which is consistent with the scope of the journal. The authors' answers clarified my previous doubts.

Validity of the findings

The authors' corrections and replies have improved the article in this respect.

Reviewer 3 ·

Basic reporting

I consider the second revised version of the manuscript sufficient for acceptance.

Experimental design

I consider the second revised version of the manuscript sufficient for acceptance.

Validity of the findings

I consider the second revised version of the manuscript sufficient for acceptance.

Additional comments

I consider the second revised version of the manuscript sufficient for acceptance.